

# Comparative meta-analysis of cold snare polypectomy and endoscopic mucosal resection for colorectal polyps: assessing efficacy and safety

Shouqi Wang[1,*], Qi Zhang[1,2,*], Li Rong Meng[2], Ying Wu[1], Pedro Fong[2] and Weixia Zhou[1]

[1] The Second Affiliated Hospital, Soochow University, Soochow, China
[2] Faculty of Health Sciences and Sports, Macao Polytechnic University, Macao, China
* These authors contributed equally to this work.

Corresponding authors
Weixia Zhou,
wangxm222222@sina.com
Pedro Fong,
pedrofong@mpu.edu.mo

## ABSTRACT

Colorectal polyps are commonly treated with surgical procedures, with cold snare polypectomy (CSP) and endoscopic mucosal resection (EMR) being the two most prevalent techniques. This meta-analysis (PROSPERO ID: CRD42022336152) aimed to compare the efficacy and safety of CSP and EMR in the management of colorectal polyps. Comprehensive searches were conducted in PubMed, Embase, CINAHL, Web of Science, and Cochrane Library databases, covering publications up until June 2024. The primary outcome was complete resection rate, and secondary outcomes included en bloc resection rate, immediate and delayed bleeding, perforation, and procedure time. The Mantel–Haenszel method was employed for the analysis of binary endpoints, while the inverse variance method was used for continuous outcomes. Subgroup analysis was performed to explore potential sources of heterogeneity. Six studies involving 15,296 patients and 17,971 polyps were included in the meta-analysis. CSP had a significantly lower complete resection rate compared to EMR (OR: 0.44, 95% CI [0.21–0.94], $p = 0.0334$). However, there was no significant difference between CSP and EMR in en bloc resection rate, perforation, or procedure time. Interestingly, CSP had a significantly lower delayed bleeding rate compared to EMR (OR: 0.45, 95% CI [0.27–0.77], $p = 0.0034$), but there was no significant difference in immediate bleeding rate. In conclusion, CSP is a safe, efficient, and effective technique comparable to EMR. The choice of technique should be based on the individual patient and polyp characteristics.

# INTRODUCTION

Colorectal cancer (CRC) is the third most diagnosed cancer and the second leading cause of cancer-related death worldwide. In 2020, there were approximately 1.9 million new cases and 935,000 deaths from CRC (*Sung et al., 2021*). Colonic polyps can progress to cancer (*Gong et al., 2023*), and studies have demonstrated that endoscopic removal of the polyps can significantly decrease morbidity and mortality associated with colorectal malignancies (*Zauber et al., 2012*). Polyp removal can be performed using a variety of

techniques, each offering distinct advantages and disadvantages, thereby necessitating a thorough comparative analysis. Gastroenterologists typically select a technique based on local healthcare policies and their clinical expertise. The common techniques are cold snare polypectomy (CSP), hot snare polypectomy (HSP) and endoscopic mucosal resection (EMR).

CSP is generally considered the safer, quicker, and more cost-effective option compared to HSP and EMR, as it does not require the use of more invasive procedures, such as electrocautery or submucosal injection (*Ferlitsch et al., 2017*; *Schett et al., 2017*; *Kawamura et al., 2018*). CSP has been shown to have a lower incidence of adverse events, such as postoperative bleeding and perforation, compared to endoscopic electrocautery resection (*Uraoka et al., 2022*). However, the Guidelines for Colorectal Cold Polypectomy demonstrated this evidence is low-quality (*Uraoka et al., 2022*). The European Society of Gastrointestinal Endoscopy (ESGE) Clinical Guideline also provides low-quality evidence recommending that piecemeal CSP may reduce the risk of deep mural injury for polypectomy of sessile polyps (10–19 mm) under certain conditions (*Ferlitsch et al., 2017*). Therefore, further high-quality studies are needed to confirm the clinical advantages of using CSP.

The choice of colorectal polyp removal technique depends on the size of the lesion. For lesions smaller than 10 mm, CSP is the recommended technique (*Uraoka et al., 2022*). Its complete resection rate is comparable to HSP (*Qu et al., 2019*), but its incomplete resection rate is higher than EMR (*Zhang et al., 2018*). For lesions beyond 10 mm, the choice of technique is based on the gastroenterologist's judgment (*Uraoka et al., 2022*). To date, no definitive conclusion has been reached regarding the superiority of CSP over EMR. Therefore, we conducted this systematic review and meta-analysis to compare the efficacy and safety of CSP and EMR for colorectal polyps.

## MATERIALS AND METHODS

The detailed protocol of this systematic review and meta-analysis was prospectively registered with the International Prospective Register of Systematic Reviews (PROSPERO) under ID: CRD42022336152 to ensure its integrity and accountability.

### Search strategy

A comprehensive literature search was conducted in five databases: Cochrane Library, CINAHL, Embase, PubMed, and Web of Science. The search included articles published up to June 2024 that reported on the safety and efficacy of CSP and EMR for removing colon polyps. The search strategies were identical across all databases and can be found in Article S1.

### Selection criteria

The inclusion and exclusion criteria for this study were developed based on the Cochrane Collaboration guidelines and Preferred Reporting Items for Systematic Reviews and Meta-Analyses (PRISMA 2020) recommendations (*Page et al., 2021*). We employed the

Population, Intervention, Comparison, and Outcome (PICO) framework to define the criteria and identify the relevant studies for this review.

The inclusion criteria encompassed the following: (1) studies that included patients who had undergone colonoscopy; (2) studies assessing the effectiveness (including complete resection rate and en bloc resection rate) and safety (such as immediate bleeding, delayed bleeding, perforation, and procedure time) of CSP and EMR; and (3) randomized controlled trials (RCTs) and cohort trials with valid data.

The exclusion criteria were animal studies, conference abstracts, case reports, review articles, editorials, notes, and letters, and case series with fewer than 10 patients were excluded. Additionally, studies involving patients with inflammatory bowel disease, familial polyposis, significant infectious disease, pregnancy, chronic kidney disease, or a history of liver cirrhosis were excluded. Non-English language articles were also excluded.

The definitions of CSP and EMR may vary slightly in the literature. In this review, CSP is defined as the use of a snare alone without electrocautery. The timing for CSP varies based on polyp size, with smaller polyps (<10 mm) typically being resected immediately upon visualization. For larger polyps, careful snare positioning is essential to ensure complete excision while minimizing tissue trauma. EMR involves a submucosal injection around the polyp to achieve sufficient tissue elevation, creating a cushion that facilitates resection. After lifting, an open snare is placed around the polyp and tightened to include approximately 2–3 mm of normal mucosa at the base, followed by excision using electrocautery. This procedure allows for the removal of larger polyps in either an en bloc or piecemeal manner, depending on the polyp's size and morphology. Studies that do not adhere to these definitions were excluded from this review.

## Instrumentation and equipment

CSP is a specialized cold snare that was utilized without electrocautery. These cold snares, typically featuring a smaller diameter wire, were preferred to minimize tissue trauma. The specific model and wire thickness varied according to each study's protocol. By enabling mechanical cutting without applying thermal energy, these cold snares help reduce the risk of tissue injury and delayed bleeding.

Endoscopic mucosal resection (EMR) involved the use of a standard endoscopic submucosal injection needle for injecting a saline solution around the polyp, which was essential for achieving optimal tissue elevation. This injection usually consisted of a mixture of saline, epinephrine, and indigo carmine dye, creating a well-defined cushion that facilitates safer and more complete resection. Additionally, electrocautery-enabled snares were employed during EMR to ensure thorough excision of the lesion.

## Study selection

All search results were imported into EndNote X9 software. Duplicates were removed using the automated "find duplicates" feature, followed by a manual review. Two independent authors (Shouqi Wang and Qi Zhang) then screened the titles and abstracts to identify potential studies. Full-text articles were retrieved for further evaluation, and studies meeting the predefined eligibility criteria were selected. In cases of disagreement, a

third author (Weixia Zhou) conducted a blinded reassessment without knowledge of the other authors' opinions. Subsequently, the three authors engaged in discussions to reach a consensus. The review process was not blinded to authors, institutions, or journals. All studies comparing the effects and adverse events of CSP with EMR were included, regardless of whether the data were primary or secondary endpoints.

## Study outcomes

The primary outcome of this study is the complete resection rate, defined as the proportion of polyps that are entirely removed, with no residual tissue observed in histological examination. Complete resection is achieved when both the horizontal and vertical margins of the resected specimen are histologically negative, indicating full removal of the polyp.

Secondary outcomes include several safety and efficacy parameters. The en bloc resection rate represents the percentage of polyps successfully removed in a single piece, as assessed visually by the endoscopist, providing insight into the effectiveness of the technique in achieving complete eradication. Immediate bleeding refers to any bleeding that occurs during or immediately after the procedure, potentially necessitating additional interventions such as cauterization or clipping. Delayed bleeding includes haemorrhage requiring endoscopic intervention within 2 weeks after polypectomy, reflecting potential complications related to the healing process. Additionally, any accidental perforation of the colonic wall observed during or after the procedure is recorded, indicating potential risks associated with each technique. Finally, the procedure time is defined as the total duration required for resection, excluding anaesthesia and preparation time, which reflects the procedural efficiency and feasibility in clinical practice. Together, these outcomes will provide a comprehensive evaluation of the safety and efficacy of the techniques employed in this study.

## Data extraction

Two authors independently extracted data using standardized forms. The following data were extracted from each included study: author and year of publication, study design, country, study period, number of included patients, age and sex of patients, number of lesions, mean lesion size, morphology of the lesion, polyp removal method, complete resection rate, en bloc resection rate, immediate and delayed bleeding, perforation, and procedure time.

## Quality assessment

The methodological quality and risk of bias of each study were assessed by all authors according to the Cochrane Handbook for Systematic Reviews of Interventions version 6.4. The Cochrane Risk of Bias 2 (ROB2) (*Sterne et al., 2019*) was used to assess the quality of randomized controlled trials (RCTs), and the Risk of Bias in Non-Randomized Studies of Interventions (ROBINS-I) tool was used to assess the quality of retrospective cohort studies (*Sterne et al., 2016*). Disagreements were resolved through discussion among all authors.

## Statistical analysis

Meta-analyses were conducted using R (version 4.2.1; *R Core Team, 2022*) with the 'meta' and 'metafor' packages (*Lortie & Filazzola, 2020*). Odds ratios (ORs) along with 95% confidence intervals (CIs) were calculated using the Mantel–Haenszel method to analyze binary outcome data. Standardized mean differences (SMDs) with 95% CIs were calculated using inverse variance weighting for continuous outcomes.

Heterogeneity was assessed using the $I^2$ statistic, where values above 50% or a chi-square test *p*-value < 0.1 indicated significant heterogeneity (*Higgins & Thompson, 2002*). Potential sources of heterogeneity were investigated through sensitivity analyses, including fixed-effects *versus* random-effects models, subgroup analyses, meta-regression, and by excluding individual studies one at a time. Publication bias was assessed using a funnel plot and Egger's test, with a *p*-value < 0.1 for Egger's test considered statistically significant (*Shi et al., 2017*).

# RESULTS

## Studies selected for analysis

The research strategies and selection criteria identified 1,048 publications, primarily sourced from Embase and Web of Science (Fig. 1). After removing duplicates, 667 studies remained. Screening of titles and abstracts resulted in 50 potentially relevant articles. Upon a thorough full-text review, 44 publications were excluded, leaving six studies eligible for inclusion in the meta-analysis. These comprised three RCTs (*Zhang et al., 2018*; *Li et al., 2020*; *Rex et al., 2022*) and three cohort studies (Fig. 1) (*Noda et al., 2016*; *Ito et al., 2018*; *Saito et al., 2022*).

## Study characteristics

This meta-analysis included a total of 15,296 patients and 17,971 polyps across six selected studies. One study compared complications associated with resection treatments for colorectal polyps, including cold forceps polypectomy procedures, cold snare polypectomy (CSP), hot biopsies, hot snare polypectomy (HSP), endoscopic or piecemeal endoscopic mucosal resection (EMR/p-EMR), and endoscopic submucosal dissection (ESD) (*Saito et al., 2022*). Another trial conducted a three-arm comparison of efficacy and safety among CSP, cold snare endoscopic mucosal resection (CS-EMR), and EMR (*Li et al., 2020*). In a separate study, the efficacy of CSP and CS-EMR *versus* HSP and hot snare endoscopic mucosal resection (HS-EMR) was investigated for colorectal lesions measuring 6–15 mm (*Rex et al., 2022*). The other three studies employed a two-arm design to compare CSP and EMR, with a primary focus on their efficacy and adverse reactions (*Noda et al., 2016*; *Zhang et al., 2018*; *Ito et al., 2018*).

The six included studies were published between 2016 and 2022, with three conducted in Japan, two in China, and one in the United States (Table 1). Regarding the macroscopic type of polyps, three studies did not impose any restrictions, one excluded pedunculated polyps, and two specifically focused on nonpedunculated colorectal lesions. The mean or median age of patients treated with CSP ranged from 51.6 to 72.0 years, while for patients treated with EMR, it ranged from 51.6 to 68.0 years. Male patients accounted for 55% to
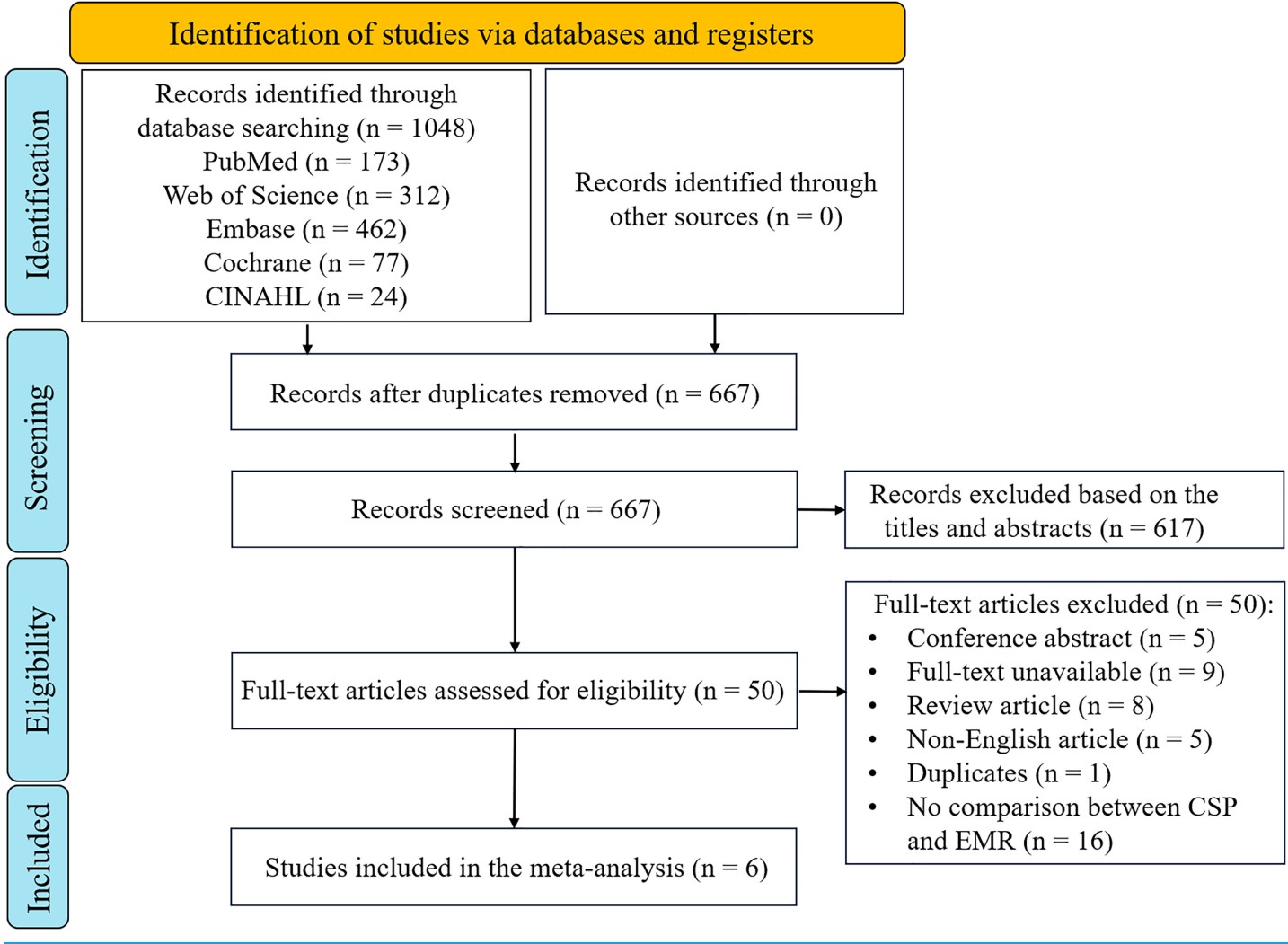

**Figure 1 PRISMA-guided assessment protocol for studies identified in the meta-analysis.** CSP, cold snare polypectomy; EMR, endoscopic mucosal resection.

78.3% of cases in both CSP and EMR groups. The average polyp size ranged from 5.00 to 11.95 mm for CSP and from 4.00 to 12.22 mm for EMR. Five of the six studies reported the complete resection rate, with CSP groups ranging from 42.9% to 100.0% and EMR groups ranging from 60.9% to 98.5%. Three studies provided data on the en bloc rate. The incidences of perforation, immediate bleeding, and delayed bleeding were low. The duration of the procedures reported in the three RCTS varied between 2.12 and 4.70 min for the CSP groups and between 3.41 and 5.50 min for the EMR groups (Tables 1 and 2).

## Assessment of inherent bias

According to the Cochrane risk of bias analysis, the overall risk of bias for the included studies was considered to be low to moderate. However, one RCT failed to report the blinding of participants and personnel (Fig. 2). Another RCT by *Zhang et al. (2018)* did not adequately describe the randomization method, which may have compromised allocation

**Table 1 Baseline characteristics of included studies in the meta-analysis.**

| First author | Publication year | Study type | Location | Time interval | Polyp size (mm) | Macroscopic type |
|---|---|---|---|---|---|---|
| Hisatsugu Noda | 2016 | Cohort study | Japan | May 2014–June 2015 | 3–15 | Unlimited |
| Qisheng Zhang | 2018 | RCT | China | March 2014–May 2016 | 6–9 | Non-pedunculated colorectal lesions |
| Akihiro Ito | 2018 | Cohort study | Japan | September 2014–October 2016 | ≤9 | Non-pedunculated colorectal lesions |
| Dazhou Li | 2020 | RCT | China | July 2017–March 2019 | 6–20 | Unlimited |
| Douglas Rex | 2022 | RCT | US | August 2018–March 2021 | 6–15 | Non-pedunculated colorectal lesions |
| Yutaka Saito | 2022 | Cohort study | Japan | January 2015–March 2017 | Unlimited | Unlimited |

concealment. On the other hand, the ROBINS-I tool indicated that although these studies employed methodological controls, some risks of bias remained, particularly in the classification of interventions and reporting of outcomes (Fig. 3; Table 3).

In this meta-analysis, we assessed the risk of bias for each study, identifying specific studies with higher risks and the reasons for these assessments. For the RCTs, *Zhang et al. (2018)* presented an unclear risk in the randomization process due to insufficient description, which could lead to allocation concealment issues. In contrast, *Li et al. (2020)* implemented appropriate measures across all assessed domains, including randomization, deviations from intended interventions, and outcome measurement, resulting in a low-risk rating for all domains. *Rex et al. (2022)* also showed an unclear risk in the randomization process and reporting of results, primarily due to a lack of detail in randomization and outcome selection, which may lead to selective reporting bias.

Regarding retrospective cohort studies, *Noda et al. (2016)* were rated as low risk across all domains, including confounding and selection bias, indicating an overall low risk of bias. Conversely, *Ito et al. (2018)* exhibited moderate risk in several bias domains, such as confounding, participant selection, deviations from intended interventions, missing data, and selection of reported results. These issues likely stemmed from the retrospective design and inconsistencies in intervention details, which hindered adequate control of confounders and participant selection. Similarly, *Saito et al. (2022)* were rated with moderate risk in confounding, participant selection, deviations from intended interventions, and selection of reported results, as the limitations of retrospective data made it challenging to fully control confounding factors. Potential selective reporting bias and inconsistencies in participant selection and intervention details further impacted the reliability of this study.

Table 2 Characteristics of included studies comparing cold snare polypectomy (CSP) and endoscopic mucosal resection (EMR) for colon polyp resection.

| First author | Number of lesions | Number of patients (male/total) | Mean patient age (year) | Mean polyp size (mm) | Complete resection rate | En bloc rate | Perforation rate | Immediate bleeding | Delayed bleeding | Procedure time (min) |
|---|---|---|---|---|---|---|---|---|---|---|
| Hisatsugu Noda | | | | | | | | | | |
| CSP | 175 | 83/106 | 66.8 | 5.0 | 93/175 | NA | 0/175 | NA | 0/175 | NA |
| EMR | 1,010 | 285/423 | 67.9 | 6.8 | 615/1,010 | NA | 0/1,010 | NA | 2/1,010 | NA |
| Qisheng Zhang | | | | | | | | | | |
| CSP | 267 | 96/179 | 64.5 ± 7.7 | 7.4 ± 1.5 | 194/212 | 234/267 | NA | 5/267 | 0/267 | 4.7 ± 3.4 |
| EMR | 258 | 101/179 | 65.8 ± 9.4 | 7.7 ± 1.4 | 200/203 | 245/258 | NA | 3/258 | 0/258 | 5.5 ± 2.7 |
| Akihiro Ito | | | | | | | | | | |
| CSP | 373 | 85/126 | 72 | | (median) | 4 | (median) | 79/184 | NA | 0/373 |
| NA | 2/373 | NA | | | | | | | | |
| EMR | 699 | 261/408 | 68 | | (median) | 5 | (median) | 114/184 | NA | 0/699 |
| NA | 19/699 | NA | | | | | | | | |
| Dazhou Li | | | | | | | | | | |
| CSP | 244 | 77/129 | 51.63 ± 14.395 | 11.95 ± 3.35 | 199/244 | 139/140 | 1/244 | 23/244 | 3/244 | 3.01 ± 1.019 |
| EMR | 267 | 80/137 | 51.59 ± 14.495 | 12.22 ± 3.77 | 255/267 | 230/236 | 2/267 | 5/267 | 7/267 | 3.41 ± 0.925 |
| Douglas Rex | | | | | | | | | | |
| CSP | 68 | 34/59 | 66.2 ± 9.9 | 9.4 ± 3.1 | 68/68 | 40/68 | NA | 0/59 | NA | 2.12 ± 1.52 |
| EMR | 65 | 39/56 | 67.0 ± 8.4 | 10.0 ± 3.1 | 125/136 | 55/65 | NA | 4/56 | NA | 5.12 ± 2.53 |
| Yutaka Saito | | | | | | | | | | |
| CSP | 4,770 | 3,218/4,437 | 67.13 ± 10.75 | 5.68 | NA | NA | 2/4,770 | 7/4,770 | 12/4,770 | NA |
| EMR | 9,775 | 6,499/9,057 | 65.47 ± 11.58 | 9.82 | NA | NA | 6/9,770 | 31/9,775 | 48/9,775 | NA |

## Complete resection rate

Five studies reported the complete resection rate for lesions removed by CSP and EMR (*Noda et al., 2016*; *Zhang et al., 2018*; *Ito et al., 2018*; *Li et al., 2020*; *Rex et al., 2022*), encompassing a total of 883 lesions removed by CSP and 1,800 lesions removed by EMR. The complete resection rate ranged from 42.9% to 100% in the CSP group and 60.9% to 98.5% in the EMR group (Table 2). A significant difference in the complete resection rate was observed between CSP and EMR (OR: 0.44, 95% CI [0.21–0.94], $p = 0.0334$), with high heterogeneity ($I^2 = 80\%$).

Subgroup analyses were conducted based on the study location (China, Japan, and the U.S.), significant differences between CSP and EMR in the Japanese and Chinese analyses (OR: 0.61, 95% CI [0.48–0.79], $p < 0.001$ and OR 0.20, 95% CI [0.11–0.35], $p < 0.001$, respectively) (Fig. 4A). The Chinese analysis exhibited no heterogeneity ($I^2 = 0\%$), while

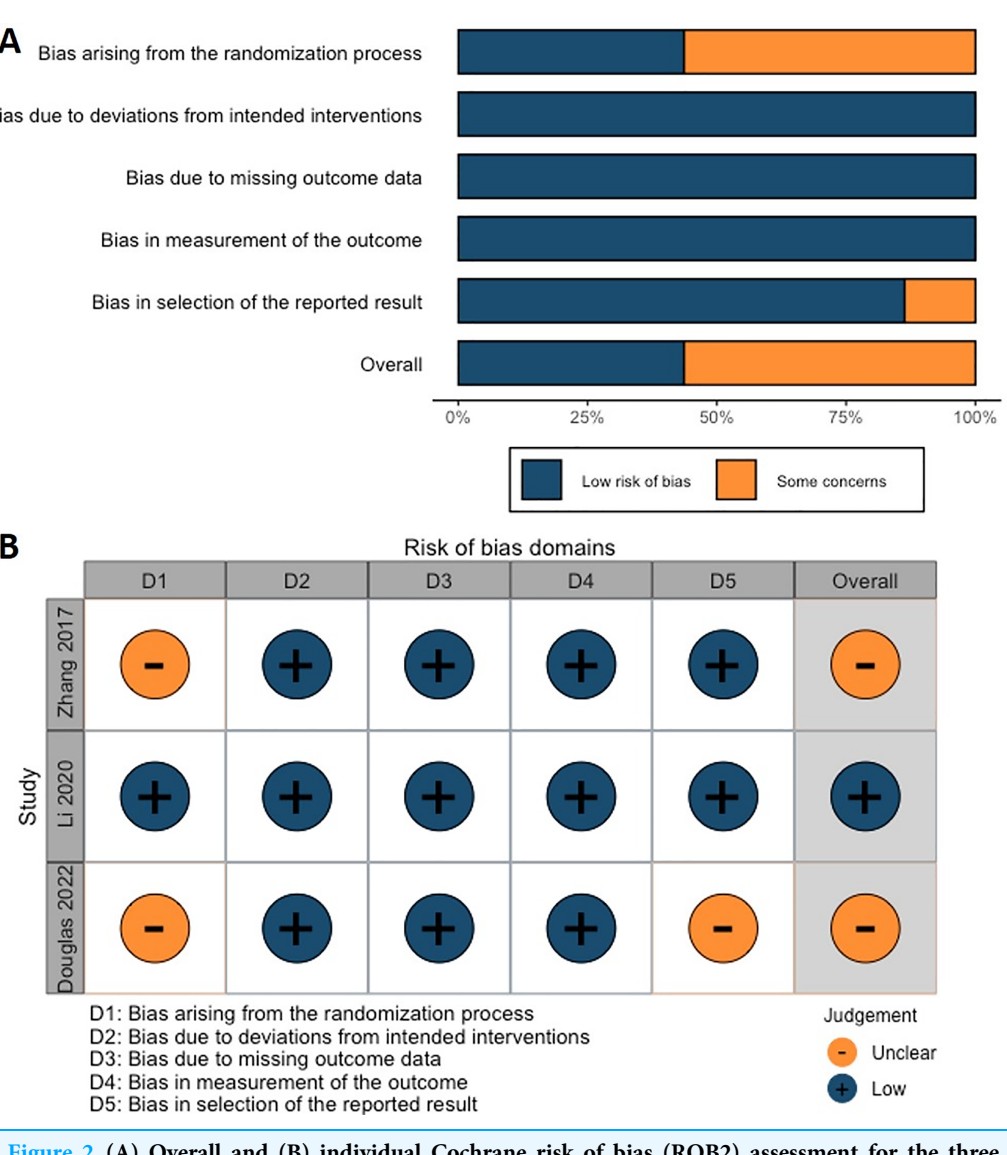

**Figure 2** (A) Overall and (B) individual Cochrane risk of bias (ROB2) assessment for the three included RCTs studies.

the Japanese analysis demonstrated moderate heterogeneity ($I^2 = 65\%$). Due to the limited number of included studies, a comprehensive assessment of publication bias was challenging.

## En bloc resection rate

This analysis included the three studies that examined the en bloc resection rate (Table 2), with a total of 475 lesions in the CSP group and 559 lesions in the EMR group. En bloc resection rates ranged from 58.8% to 99.3% for CSP and 84.6% to 97.5% for EMR. Statistical analysis revealed no significant difference in en bloc resection rates between CSP and EMR (OR: 0.48, 95% CI [0.16–1.43], $p = 0.1872$). However, there was moderate heterogeneity ($I^2 = 61\%$) (Fig. 4B).

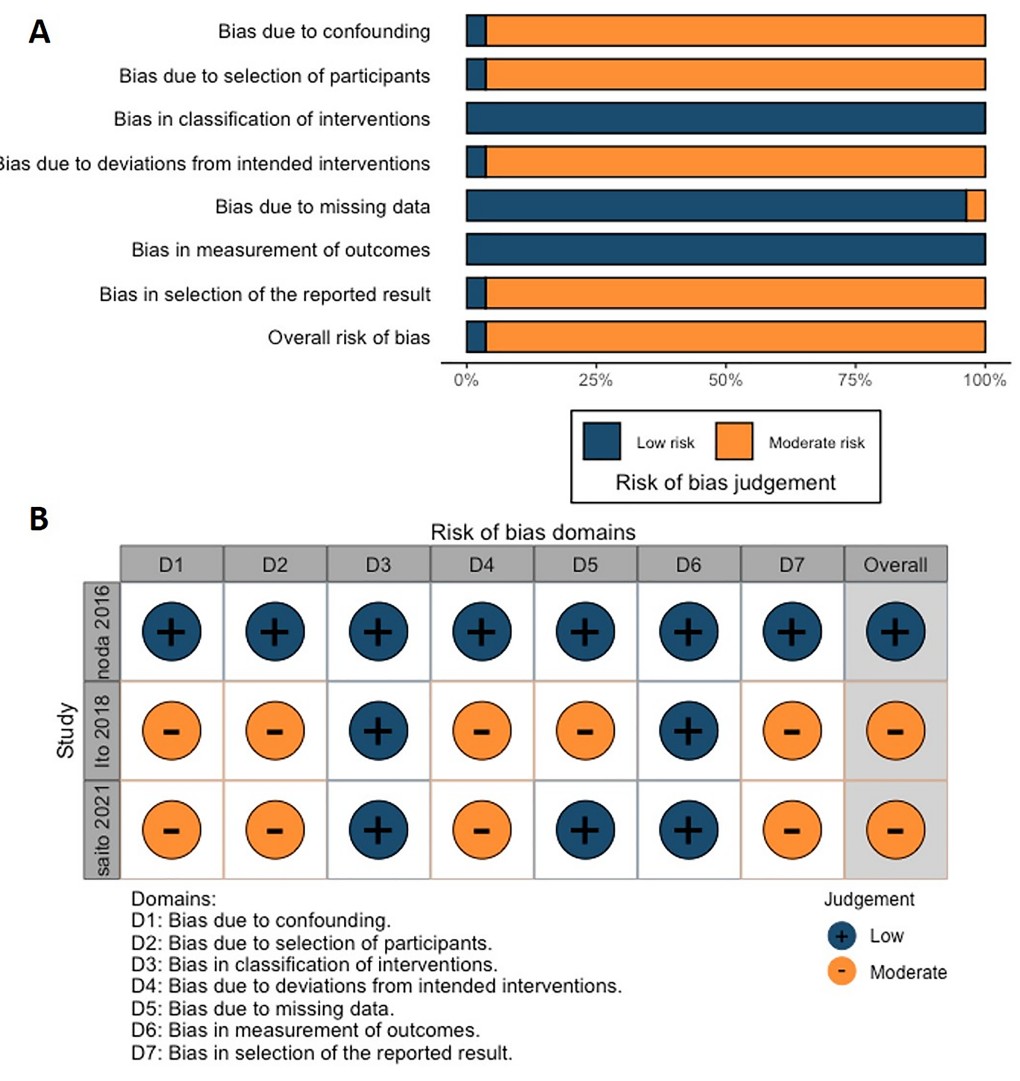

**Figure 3** (A) Overall and (B) individual Risk of Bias in Non-Randomized Studies of Interventions (ROBINS-I) assessment for the three included retrospective cohort studies.

## Perforation

Perforation was reported in four out of the six included studies (Table 2), involving a total of 5,262 lesions removed in the CSP group and 11,746 lesions in the EMR group. No statistical significance was found between CSP and EMR (OR: 0.67, 95% CI [0.18–2.53]; $p = 0.5525$), and no heterogeneity was observed ($I^2 = 0\%$) (Fig. 5A).

## Immediate bleeding

Immediate bleeding was reported in four studies (Table 2). Subgroup analyses based on study location (China, Japan, and the U.S.) were conducted due to significant heterogeneity ($p = 0.9513$, $I^2 = 83\%$). A statistically significant difference was found between CSP and EMR in the Chinese analysis (OR: 3.35, 95% CI [1.05–10.74], $p = 0.0087$)

**Table 3 Newcastle-Ottawa Scale (NOS) scores of the included cohort studies.**

| NOS items | Studies | | |
| --- | --- | --- | --- |
| | Hisatsugu Noda | Akihiro Ito | Yutaka Saito |
| Selection | | | |
| *Representativeness of the exposed cohort* | * | * | * |
| *Selection of the non-exposed cohort* | * | * | * |
| *Ascertainment of exposure* | * | * | * |
| *Demonstration that outcome of interest was not present at the start of the study* | * | * | * |
| Comparability of cohorts based on the design or analysis | ** | ** | ** |
| Outcome | | | |
| *Assessment of outcome* | * | * | * |
| *Follow-up long enough for outcomes to occur* | * | * | * |
| *Adequacy of follow-up of cohorts* | 0 | 0 | 0 |
| Total scores (9/9) | 8/9 | 8/9 | 8/9 |

**Note:**
Each asterisk (*) represents a point awarded for meeting specific criteria within three domains: selection, comparability, and outcome. Each domain has its criteria, and studies can score a maximum of one star for each criterion they meet, with the exception of comparability, which can have a maximum of two stars. The total possible score is nine stars. A higher score indicates a higher quality of the study.

(Fig. 5B), with low heterogeneity ($I^2$ = 46%). However, the overall analysis indicated that CSP may not differ significantly from EMR (OR: 1.06, 95% CI [0.23–4.68], $p$ = 0.9513).

### Delayed bleeding

Five studies with data on delayed bleeding were included in this analysis, comprising 5,829 patients in the CSP group and 12,009 patients in the EMR group (Table 2). One of these studies reported no delayed bleeding events. For the remaining studies, the delayed bleeding rate ranged from 0.00% to 1.23% in the CSP group and 0.20% to 2.71% in the EMR group. The CSP group had a significantly lower delayed bleeding rate compared to that of EMR (OR: 0.45, 95% CI [0.27–0.77], $p$ = 0.0034), with no heterogeneity observed ($I^2$ = 0%) (Fig. 5C).

### Procedure time

Three out of the six studies included in this analysis were RCTs that measured procedure time (Table 1). These studies involved a total of 367 patients in the CSP group and 372 patients in the EMR group (Table 2). These studies exhibited a high level of heterogeneity ($I^2$ = 92%), leading to a subgroup analysis based on the geographic location of the studies (China *vs.* U.S.) (Fig. 5D). The subgroup analysis revealed that the CSP group in China had a significantly shorter procedure duration compared to the EMR group (SMD −0.32, 95% CI [−0.48 to −0.17], $p$ < 0.0001) (Fig. 5D), and no heterogeneity was observed ($I^2$ = 0%). However, when considering the overall effect, the analysis indicated that CSP may not differ significantly from EMR (SMD −0.68, 95% CI [−1.39 to 0.02], $p$ = 0.0580).

### DISCUSSION

The findings of this meta-analysis indicated that significant differences were found between CSP and EMR regarding the complete resection rate and delayed bleeding, but not

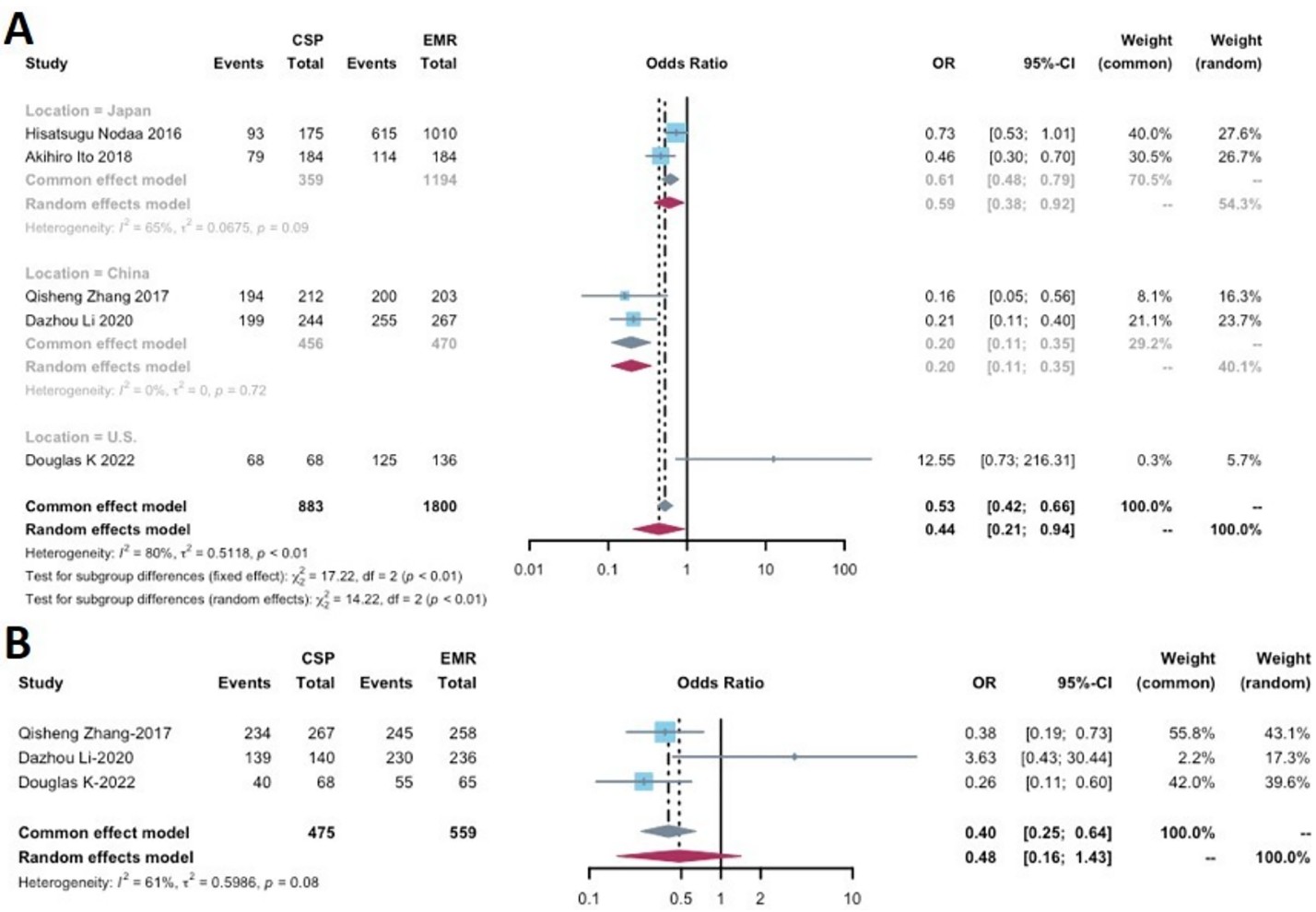

**Figure 4 Forest plot comparing the effectiveness measurements of cold snare polypectomy (CSP) and endoscopic mucosal resection (EMR) for (A) complete resection rate and (B) en bloc rate.**

in other measurements, including the en bloc resection rate, perforation, immediate bleeding, and the procedure time.

To date, there have been limited studies directly comparing the efficacy and safety of CSP and EMR for treating colorectal polyps. Previous investigations have primarily focused on comparing EMR with endoscopic submucosal dissection (ESD) (*Chao, Zhang & Si, 2016*; *Pan et al., 2018*; *Shahini et al., 2022*) or CSP with hot snare polypectomy (HSP) (*Fujiya et al., 2016*; *Takeuchi et al., 2022*). For instance, *Yuan et al. (2021)* conducted a systematic review and pooled analysis comparing the effectiveness and safety of different endoscopic resection methods. However, their study only included the R0 resection rate and en bloc resection rate as outcome measures, with limited data available for CSP. In contrast, our meta-analysis in this study provides more specific comparisons of polyp removal methods and includes multiple outcome measures for direct comparisons. We also introduced the assessment of the complete resection rate and delayed bleeding between CSP and EMR to our analysis, providing clinically valuable results.

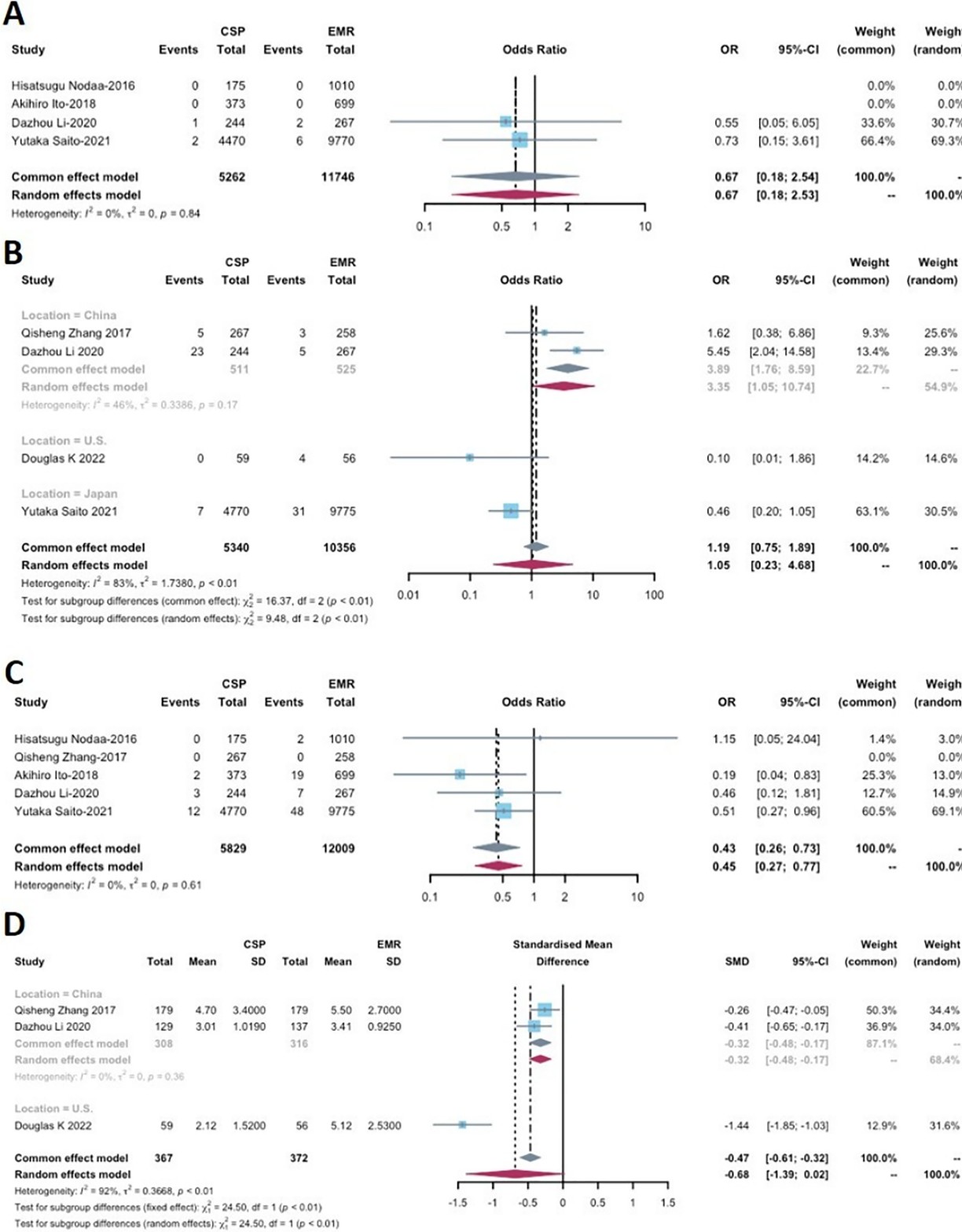

**Figure 5 Forest plot comparing the safety profiles of cold snare polypectomy (CSP) and endoscopic mucosal resection (EMR) for (A) perforation, (B) immediate bleeding, (C) delayed bleeding and (D) procedure time.**

## Comparative evaluation of safety

Consistent with a previous study (*Abe et al., 2018*), our findings demonstrate that CSP does not significantly reduce the incidence of perforation compared to other polyp resection approaches. The incidence of perforation in clinical studies is rare for both CSP and EMR (*Abe et al., 2018*), which may explain the lack of difference in perforation rates between the two methods in our study.

The comparable low incidence of immediate bleeding between CSP and EMR may be attributed to a similar low incidence factor (*Bahin et al., 2016*), However, in terms of delayed bleeding, CSP is generally associated with a low incidence rate, as supported by previous studies (*Paspatis et al., 2011*; *Bahin et al., 2016*; *Chang et al., 2019*). Our findings reveal a significantly lower incidence of delayed bleeding with CSP compared to EMR. This difference can be explained by the absence of electrocautery in CSP, which eliminates the potential thermal injury to the colonic wall and lowers the risk of subsequent delayed bleeding (*Lorenzo-Zúñiga et al., 2014*). Additionally, the mechanical tearing wounds created by cold polypectomy may facilitate easier healing compared to the thermal tissue damage induced by electrocautery at the edges of the mucosal defect in EMR (*Lorenzo-Zúñiga et al., 2014*). The thermal injury can lead to tissue necrosis that extends both horizontally and vertically, potentially increasing the likelihood of delayed bleeding.

The size of the polyp and the use of electrocautery in EMR can significantly influence its safety. For instance, *Guo, Li & Zhu (2022)* conducted a study comparing Cold Snare Endoscopic Mucosal Resection (CS-EMR) and Hot Snare Endoscopic Mucosal Resection (HS-EMR) in the treatment of colorectal polyps measuring 6–9 mm. Their findings indicated that the HS-EMR group experienced a higher incidence of post-polypectomy syndrome and longer hospital stays. This suggests that non-electrocautery removal with cold snares may provide greater safety for smaller polyps. Similarly, a multicenter study by *Mangira et al. (2023)* examined the safety and efficacy of CSP and C-EMR for non-pedunculated colonic polyps measuring 10–19 mm. The results demonstrated that both CSP and C-EMR had low rates of incomplete resection and recurrence, as well as a low incidence of complications. These findings align with the results of this meta-analysis, further indicating that CSP may be a safer option for the removal of polyps of varying sizes.

It is important to note that other factors may also have influenced the findings of this study. For instance, studies have suggested that the use of more endoscopic clips may reduce the rate of delayed bleeding (*Matsumoto et al., 2016*; *Spadaccini et al., 2020*). However, the prophylactic use of hemostatic clips has not been proven to prevent delayed bleeding after conventional polypectomy (*Matsumoto et al., 2016*; *Spadaccini et al., 2020*). Furthermore, the number of endoscopic clips used was not specified in the included studies of this meta-analysis.

In terms of procedure time, we observed a non-significant difference, with CSP being slightly shorter than EMR (Table 2; Fig. 5D). One possible explanation for this finding is the additional time required in EMR for the elevation of flat polyps before submucosal injection (*Ichise et al., 2011*; *Uraoka et al., 2014*). However, it is also possible that the flat polyps can be more difficult to snare in CSP and lead to a longer procedure time.

## Comparative evaluation of effectiveness

In this study, we observed comparable en bloc resection rates between CSP and EMR (Table 2; Fig. 4B). Previous literature has indicated that the rate of en bloc resection tends to be higher for specific lesions, such as residual or recurrent adenoma (RRA) in nonpedunculated colorectal lesions (*Belderbos et al., 2014*). However, our study included a diverse range of polyps, which may have contributed to the lack of difference in en bloc resection rates between CSP and EMR.

The EMR has demonstrated enhanced complete resection than CSP in this meta-analysis. Several factors may contribute to this difference. EMR allows for the removal of larger lesions using a "piecemeal" resection technique, where polyps are removed in multiple fragments (*Scheer et al., 2022*). Additionally, EMR offers the option for additional therapeutic interventions, such as submucosal injection of agents, thermal hemostasis, or the use of endoscopic clips (*Zhang et al., 2018*). EMR is commonly employed in polyp removal using a snare and an electrosurgical unit to ensure complete resection (*Ferlitsch et al., 2017*).

Furthermore, the studies included in our analysis focused on lesions with a diameter of less than 20 mm. Although CSP may be applied to larger polyps (*Tate et al., 2018*; *Yoshida et al., 2021*), Japanese guidelines still recommend its use for adenomas smaller than 10 mm (*Uraoka et al., 2022*). A study has shown that the complete resection rates for CSP and EMR are similar for colorectal polyps measuring 3–10 mm (*Wang et al., 2024*). Therefore, using CSP in these relatively large lesions might have contributed to the lower complete resection rate compared to EMR.

In addition, the study by *King et al. (2021)* assessed the en bloc resection rates of EMR for polyps measuring 20 mm or smaller. The findings indicated that polyp size and the number of procedures performed are significant predictors of successful en bloc resection. In general, larger polyps and those located in the right colon were associated with an increased risk of resection failure.

The utilization of various types of CSP procedures and adherence to clinical guidelines vary among endoscopists with varying levels of training (*Torres et al., 2022*). The skill and experience of the endoscopists have also been shown to impact the rate of complete resection (*Pohl et al., 2013*). In this analysis, the included studies involved endoscopists who were specialists with a specific interest in colorectal cancer prevention and polypectomy, which may limit the generalizability of the results (*Rex et al., 2022*). However, it is worth noting that the studies in our analysis explicitly stated that polypectomies were performed by experienced endoscopists in both groups, and a specific number of endoscopy examinations had to be completed each year (*Zhang et al., 2018*; *Li et al., 2020*). This suggests that the variability in endoscopist skills was minimized within the study population.

The subgroup analysis of the complete resection rate in China, Japan, and the U.S. revealed heterogeneity (Fig. 4A). This heterogeneity may be attributed to several factors. One potential factor is the variation in definitions of complete resection. Among the studies analyzed, two defined complete resection as the absence of histologically negative

horizontal and vertical edges of the resected polyp, whereas three others defined it as the absence of visible adenomas or hyperplastic tissue in forceps samples histologically taken from four quadrants of tissue at the base and wound edge. Therefore, these differences may have influenced the number of complete resections reported in the included studies.

In addition to definitional discrepancies, other factors also contributed to the observed heterogeneity. Variability in polyp size is a significant factor, as this study included polyps ranging from less than 10 mm to up to 20 mm. This variation directly affects the ease of achieving complete resection and the associated risk of complications, with smaller polyps generally being easier to remove completely. CSP is generally preferred for polyps smaller than 10 mm due to its lower risk of bleeding, as it does not involve electrocautery. This makes CSP suitable for the safe removal of small lesions (*Janik, 2023*). In contrast, for polyps larger than 10 mm, EMR shows a higher complete resection rate, making it more advantageous for managing larger or more complex lesions (*Yuan et al., 2021*).

Furthermore, differences in specific techniques used for CSP and EMR, such as variations in the diameter of cold snares or the use of electrocautery, may impact the depth and effectiveness of the resection. In addition, the experience level of the operators varied between studies, with higher levels of expertise in endoscopists generally associated with better outcomes in terms of complete resection rates and complication management. These factors collectively contribute to the heterogeneity observed among studies and were further explored in our sensitivity analysis.

Previous study indicates that serrated lesions, such as sessile serrated adenomas (SSA), are associated with a higher risk of recurrence, while adenomas are generally classified as precancerous lesions (*Erichsen et al., 2016*). Consequently, EMR may be more appropriate for adenomas, as it offers a higher rate of complete removal and minimizes recurrence rates (*Pellise et al., 2017*). In contrast, for low-risk small serrated lesions, CSP may be a safer alternative (*Repici et al., 2012*). Additionally, flat or superficial lesions, such as 0-IIa and 0-IIb, can often be safely removed using CSP (*Facciorusso et al., 2015*; *Burgess et al., 2014*). For more complex lesions like 0-Is, however, EMR is typically the preferred option due to its higher complete resection rate (*Facciorusso et al., 2015*; *Burgess et al., 2014*).

Moreover, the use of narrow-band imaging (NBI) during biopsy sampling could potentially affect the complete resection rate of polyps (*Tsuji et al., 2018*). The application of NBI as a diagnostic tool may influence the accuracy of polyp detection and subsequent resection. However, the included studies in this analysis did not provide information regarding the use of NBI. Additionally, the choice of snares used in the resection procedure may also contribute to the varying rates of complete polyp resection (*Din et al., 2015*). Different types of snares could have different efficacy and safety profiles, which might impact the overall success of the resection.

## Limitations

This meta-analysis is limited by the absence of data on recurrence rates after CSP and EMR in the six included studies. Recurrence rates are crucial for prognosis, as CSP of sessile serrated lesions (SSLs) ≥10 mm has been associated with a higher recurrence rate

(*Yoshida et al., 2021*). The unavailability of this information precluded the investigation of this factor.

Additionally, this study lacks a cost-effectiveness analysis comparing CSP and EMR. CSP is generally more economical than EMR as it does not require an electrosurgical system or submucosal injections, potentially resulting in lower cumulative direct costs (*Oh, Choi & Cho, 2022*). CSP relies less on complex devices and involves simpler procedures that lead to shorter operation times, thereby reducing resource utilization. In contrast, EMR generally require higher costs due to the necessary electrosurgical equipment and additional consumables involved in the procedure. For polyps larger than 20 mm, EMR resection margins may require thermal ablation to reduce adenoma recurrence rates, which can further escalate costs through the need for submucosal injectables or postoperative thermal coagulation measures (*Chandrasekar et al., 2020*). From the perspectives of both patients and physicians, using the lower-cost and safer CSP can significantly reduce medical expenses, especially for the removal of small to medium-sized polyps. This information is especially important in resource-limited settings where managing healthcare costs is crucial.

The analysis conducted in this study did not include the thickness of the wire utilized in CSP. This decision was based on previous research findings (*Giri et al., 2022*), which indicated that the diameter of the wire is not associated with either the effectiveness or safety of CSP.

This meta-analysis includes only six studies that met the selection criteria, which may limit the statistical power of the results. Nevertheless, these studies involved a substantial sample size of 15,296 patients, and subgroup analyses were employed to address heterogeneity between studies. This study adheres to the Cochrane Collaboration's rigorous methodology, including comprehensive search strategies, independent selection and data abstraction, and quality assessment, to enhance the reliability and generalizability of the findings.

## CONCLUSIONS

This meta-analysis provides evidence that EMR has a higher complete resection rate but a higher delayed bleeding rate compared to CSP. However, en bloc resection rate and perforation rate are similar between the two techniques. Interestingly, CSP does not increase the risk of immediate bleeding or reduce the procedure time. Based on these findings, we conclude that CSP is a safe and effective polypectomy technique comparable to EMR. Further research is warranted to investigate the long-term follow-up of polyp recurrence and to conduct a cost-benefit analysis comparing CSP and EMR.

### Funding

This work was supported by the Suzhou Nursing Association GuSu nursing talent "Qingmiao" plan (SHQM202304). The funders had no role in study design, data collection and analysis, decision to publish, or preparation of the manuscript.

## Grant Disclosures

The following grant information was disclosed by the authors:
Suzhou Nursing Association GuSu Nursing Talent "Qingmiao" Plan: SHQM202304.

## Competing Interests

The authors declare that they have no competing interests.

## Author Contributions

- Shouqi Wang conceived and designed the experiments, performed the experiments, analyzed the data, prepared figures and/or tables, authored or reviewed drafts of the article, and approved the final draft.
- Qi Zhang conceived and designed the experiments, performed the experiments, analyzed the data, prepared figures and/or tables, authored or reviewed drafts of the article, and approved the final draft.
- Li Rong Meng performed the experiments, analyzed the data, authored or reviewed drafts of the article, and approved the final draft.
- Ying Wu performed the experiments, analyzed the data, authored or reviewed drafts of the article, and approved the final draft.
- Pedro Fong conceived and designed the experiments, performed the experiments, analyzed the data, prepared figures and/or tables, authored or reviewed drafts of the article, and approved the final draft.
- Weixia Zhou conceived and designed the experiments, performed the experiments, analyzed the data, authored or reviewed drafts of the article, and approved the final draft.

## Data Availability

    This is a systematic review and meta-analysis.

## Supplemental Information

Supplemental information for this article can be found online at http://dx.doi.org/10.7717/peerj.18757#supplemental-information.

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
