# Peer review of "Comparative meta-analysis of cold snare polypectomy and endoscopic mucosal resection for colorectal polyps: assessing efficacy and safety"

_PeerJ, doi:10.7717/peerj.18757_

## Round 0.1 · original submission · Minor Revisions

The manuscript was reviewed by two referees, both of whom have suggested many revisions. Please address all of their concerns with changes to the manuscript and/or responses in a Letter of Response to Review accompanying the revised manuscript. Additionally, I suggest going through this checklist and ensuring that the manuscript content covers its points: https://els-jbs-prod-cdn.jbs.elsevierhealth.com/pb/assets/raw/Health%20Advance/journals/asmr/ASMR_Systematic_Review_and_Meta-analysis_Checklist.pdf

Reviewer 1 ·

Basic reporting

No comment

Experimental design

Some comments on your paper
- Lines 102-106: Add information on how CSP and EMR were performed, such as specific techniques, timing, and conditions of implementation.
- Clearly define the specific outcomes that the study aims to evaluate, such as primary outcomes and secondary outcomes.
- Describe the types of instruments or tools used in the study, including specific equipment like the type of snare.

Validity of the findings

Some comments on your paper
- When discussing heterogeneity and sensitivity analysis, specify the particular factors that contributed to this heterogeneity. Suggested additional information: "It would be beneficial to clarify factors such as differences in polyp size or implementation techniques that could influence this heterogeneity."
- In the risk assessment section, provide a concise summary of each study and the reasons why certain studies have higher risks. For example: "Clearly identify the studies with high risk and provide specific reasons so readers can follow along more easily."
- Expand the discussion to compare with other studies.
- Consider adding information on the costs associated with each method, such as "Discuss the treatment costs for both CSP and EMR to help physicians and patients make more informed decisions."

Additional comments

Consider further analysis of differences based on lesion size (<10 mm or >10 mm), histological characteristics (serrated lesions or adenomas), and lesion type according to the Paris classification to identify potential differences between the two methods.

·

Basic reporting

The shared raw data, tables, and figures were satisfactory. The supplementary sections, including the search strategy, were also well-presented. The overall quality and clarity of the manuscript are acceptable. However, the introduction section could benefit from further refinement, particularly when compared to the later sections of the article.

Two specific suggestions for improving the introduction:

1. On lines 47 and 48, the sentence "undergoing endoscopy and their removal can..." could be clarified by rephrasing it as "endoscopic removal of the polyps...". This alteration ensures that the pronoun "their" unambiguously refers to the polyps.

2. The sentence on line 49, "Different techniques are available for polyp removal," could be expanded to highlight the diverse range of methods used. Consider revising it to read, "Polyp removal can be performed using different techniques, each with its own advantages and disadvantages." This revised sentence will more effectively introduce the study's focus on comparing the various techniques and their respective pros and cons.

Experimental design

The study question is well-posed and holds significant interest.

Validity of the findings

no comment

---

## Round 0.2 · Minor Revisions

The revised manuscript was examined by the two referees of the original submission, and one of them has suggested a minor correction. Kindly revise accordingly.

Reviewer 1 ·

Basic reporting

No comment

Experimental design

No comment

Validity of the findings

No comment

Additional comments

The author's article has incorporated responses and provided additional updated data to further clarify the value of the article.

·

Basic reporting

The manuscript is well-documented and meets the required standards. Just needs some minor revisions.

Experimental design

no comment

Validity of the findings

no comment

Additional comments

The last sentence of the introduction “This highlights the importance of clinicians considering appendiceal mucocele in the differential diagnosis, as early identification can prevent complications.” Is not needed to be pointed out in the introduction section. Can be transferred to discussion/conclusion.
The sentence “In the initial laboratory tests, the complete blood and white blood cell counts were elevated by about 13 thousand others were normal” is not a professional one. Firstly, it does not contain scales (e.g., Per ml) and secondly, globally the test name is CBC (complete blood count) and lastly, you could have improved that by saying “Laboratory findings were unremarkable, with the exception of a mild leukocytosis.”

---

## Round 0.3 · accepted · Accept

The concerns of the referees have been satisfactorily addressed in the revised submission.